# Lumbar Puncture and Meningitis in Infants with Proven Early- or Late-Onset Sepsis: An Italian Prospective Multicenter Observational Study

**DOI:** 10.3390/microorganisms11061546

**Published:** 2023-06-10

**Authors:** Luca Bedetti, Francesca Miselli, Chiara Minotti, Giuseppe Latorre, Sabrina Loprieno, Alessandra Foglianese, Nicola Laforgia, Barbara Perrone, Matilde Ciccia, Maria Grazia Capretti, Chiara Giugno, Vittoria Rizzo, Daniele Merazzi, Silvia Fanaro, Lucia Taurino, Rita Maria Pulvirenti, Silvia Orlandini, Cinzia Auriti, Cristina Haass, Laura Ligi, Giulia Vellani, Chryssoula Tzialla, Cristina Tuoni, Daniele Santori, Mariachiara China, Lorenza Baroni, Silvia Nider, Federica Visintini, Lidia Decembrino, Giangiacomo Nicolini, Roberta Creti, Elena Pellacani, Arianna Dondi, Marcello Lanari, Belinda Benenati, Giacomo Biasucci, Lucia Gambini, Licia Lugli, Alberto Berardi

**Affiliations:** 1Neonatal Intensive Care Unit, University Hospital of Modena, 41224 Modena, Italy; luca.bedetti@unimore.it (L.B.); alberto.berardi@unimore.it (A.B.); 2PhD Program in Clinical and Experimental Medicine, University of Modena and Reggio Emilia, 21124 Modena, Italy; 3Neonatal Intensive Care Unit, Ecclesiastical General Hospital F. Miulli, 70021 Acquaviva delle Fonti, Italy; g.latorre@miulli.it; 4Department of Biomedical Science and Human Oncology (DIMO), Neonatal Intensive Care Unit, University Hospital of Bari “Aldo Moro”, 70124 Bari, Italy; 5Neonatal Intensive Care Unit, Azienda Ospedaliero Universitaria delle Marche, 60126 Ancona, Italy; 6Neonatal Intensive Care Unit, Women’s and Children’s Health Department, Maggiore Hospital, 40133 Bologna, Italy; 7Neonatal Intensive Care Unit, Women’s and Children’s Health Department, S. Orsola-Malpighi Hospital, 40138 Bologna, Italy; 8Pediatric Unit, Ospedale B. Ramazzini, 41012 Carpi, Italy; 9Neonatal Intensive Care Unit, Bufalini Hospital, 47521 Cesena, Italy; 10Division of Neonatology, “Valduce” Hospital, 22100 Como, Italy; 11Department of Medical Sciences, Pediatric Section, University Hospital, 44124 Ferrara, Italy; 12Neonatal Intensive Care Unit, Ospedali Riuniti, 71122 Foggia, Italy; 13Pediatric and Neonatal Unit, Morgagni-Pierantoni Hospital of Forlì, 47121 Forli, Italy; 14Neonatal Intensive Care Unit, Carlo Poma Hospital, 46100 Mantova, Italy; 15Neonatal Intensive Care Unit, Medical and Surgical Department of Fetus-Newborn-Infant, “Bambino Gesù” Children’s Hospital IRCCS, 00165 Rome, Italy; 16Neonatal Intensive Unit, San Pietro-Fatebenefratelli Hospital, 00168 Rome, Italy; 17Neonatal Intensive Unit, San Filippo Neri Hospital, 00135 Rome, Italy; 18Neonatal Intensive Unit, ARNAS Civico-Di Cristina-Benfratelli, 90127 Palermo, Italy; 19Neonatal and Pediatric Unit, Polo Ospedaliero Oltrepò, ASST Pavia, 27100 Pavia, Italy; 20Neonatology and Neonatal Intensive Care Unit, Department of Clinical and Experimental Medicine, University Hospital of Pisa, 56124 Pisa, Italy; 21Pediatric and Neonatal Unit, Azienda Ospedaliera Santa Maria degli Angeli, 33170 Pordenone, Italy; 22Neonatal Intensive Care Unit, Infermi Hospital, 47923 Rimini, Italy; 23Neonatal Intensive Care Unit, Santa Maria Nuova Hospital, 42123 Reggio Emilia, Italy; 24Neonatal Intensive Care Unit, Institute for Maternal and Child Health, IRCCS Burlo Garofolo, 34137 Trieste, Italy; 25Neonatology Unit, University Hospital of Udine, 33100 Udine, Italy; 26ASST Pavia, Unità Operativa di Pediatria e Nido, Ospedale Civile, 27029 Vigevano, Italy; 27Pediatric Unit, San Martino Hospital, 32100 Belluno, Italy; 28Department of Infectious Diseases, Istituto Superiore di Sanità, 00161 Rome, Italy; 29Residency in Pediatrics, Departmento of Medical and Surgical Science, University of Modena and Reggio Emilia, 41124 Modena, Italy; 30Pediatric Emergency Unit, IRCCS Azienda Ospedaliero Universitaria di Bologna, 40138 Bologna, Italy; 31Pediatric and Neonatal Unit, Women’s and Children’s Health Department, Guglielmo da Saliceto Hospital, 29121 Piacenza, Italy; 32Neonatal Intensive Care Unit, University Hospital of Parma, 43126 Parma, Italy

**Keywords:** lumbar puncture, meningitis, sepsis, newborn, prematurity, group B *streptococcus*, *Escherichia coli*

## Abstract

**Background**: To evaluate the rates of lumbar puncture (LP) in infants with culture-proven sepsis. **Study design**: We prospectively enrolled 400 infants with early- or late-onset sepsis due to Group B *streptococcus* (GBS) or *Eschericha coli*, diagnosed within 90 days of life. Rates of LP and potential variables associated with LP performance were evaluated. Moreover, cerebrospinal fluid (CSF) characteristics and results of the molecular analysis were investigated. **Results**: LP was performed in 228/400 (57.0%) infants; 123/228 LPs (53.9%) were performed after antibiotic initiation, hampering the ability to identify the pathogen in the CSF culture. However, polymerase chain reaction increased the probability of positive results of CSF analysis compared to microbiological culture (28/79, 35.4% vs. 14/79, 17.7%, *p* = 0.001). Severe clinical presentation and GBS infection were associated with higher LP rates. The rate of meningitis was 28.5% (65/228). **Conclusions**: Rates of LP are low in culture-proven neonatal sepsis and antibiotics are frequently given before LP is carried out. Thus meningitis may be underestimated, and the chances of giving an effective therapy to the newborn are reduced. LP should be performed before the start of antibiotics when there is a clinical suspicion of infection.

## 1. Introduction

Neonates are more vulnerable to meningitis than people at any other age. Neonatal meningitis affects 0.1–0.4 newborns/1000 live births, with a higher incidence in preterm infants than in neonates at term and in chronically hospitalized neonates [1]. In neonatal meningitis, case fatality is approximately 10% and up to 20–50% of surviving infants develop long-term sequelae secondary to infarcts, intracerebral thrombosis, hemorrhage, and cerebral abscesses [2,3,4]. Therefore, the early diagnosis of meningitis is essential in neonates to administer an effective treatment to reduce mortality and long-term sequelae. To date, the gold standards for the diagnosis of bacterial meningitis are microbiological culture, polymerase chain reaction, and parameters of cerebrospinal fluid (CSF), obtained through a lumbar puncture (LP) [5]. Clinical suspicion of meningitis is highest in the presence of seizures, fever, bulging fontanel, and abnormal consciousness, but in neonates and younger infants initial clinical symptoms are often subtle [6].

Although CSF analysis and culture are essential for diagnosing meningitis in infants, clinicians are sometimes reluctant to perform an LP because of the potential risks of adverse events during the procedure (i.e., hypoxia or bradycardia) or further complications (i.e., infections or bleeding) [7,8,9]. The Royal College of Obstetricians and Gynecologists recommends performing an LP if meningitis or meningococcal infection is clinically suspected, while the Committee on Fetus and Newborn of the American Academy of Pediatrics recommends an LP in suspected sepsis when meningitis cannot be clinically ruled out or there is no response to antibiotic therapy [10,11]. However, studies reporting rates of LP in the neonatal age show that it is performed far less than necessary [12], leading to the potential underestimation of cases of meningitis or the administration of unnecessary antibiotics to uninfected infants [13,14]. To maximize the accuracy of results, LP should be performed before the antibiotic administration, but some studies report that LP is frequently performed only after the blood culture has been confirmed positive, when antibiotics have already been administered [15,16]. It is unclear whether this low frequency of LP performance in neonatal sepsis depends on the clinicians or on the infant conditions themselves (e.g., premature versus term infant, severe vs. mild disease). Understanding the reasons for the differences in the execution of the LP can help to understand why in many cases it is not performed even if necessary.

The aim of this study was to evaluate the rates of LP performance in a large cohort of infants with early- (EOS) or late-onset sepsis (LOS) due to group B *streptococcus* (GBS) or *Escherichia coli* (*E. coli*), and to investigate variables associated with LP performance. 

## 2. Materials and Methods

### 2.1. Study Design

This was a prospective observational study led from 1 January 2003 to 31 December 2022. Data were derived from a surveillance network of EOS and LOS. Currently the network includes 33 centers (of which there are 13 level I–II and 20 level III units) across Northern, Central, and Southern Italy. In 2003, the network was initially launched in Emilia-Romagna, a Northern region of Italy with approximately 4,000,000 people and 35,000 live births (LBs)/year [17,18,19]. In all centers participating in the network, GBS prevention is based on antenatal screening cultures and maternal intrapartum antibiotic prophylaxis, according to the CDC guidelines [17,20,21]. Since 2003 the surveillance network concerned the active area-based surveillance of GBS infections [18], and since 2016 *E. coli* surveillance has been added [21]; GBS or *E. coli* cases (positive blood or CSF culture) occurring in an infant younger than 3 months of age are reported to the coordinating center. No exclusion criteria are used, and infants of all gestational ages were enrolled. To minimize missed cases, a reminder email is sent monthly from the coordinating center to all regional consultant pediatricians and microbiological laboratories to ask for notification. Demographics, modes of delivery, risk factors for EOS, gestational age, timing of antibiotics, blood collection and LP, CSF results, and further clinical information are obtained from the labor and delivery records by surveillance officers using a standardized form. Incomplete data are retrieved via a telephone call from the coordinating center. Because to date the surveillance network is not entirely area-based, this study had no epidemiologic purpose (i.e., incidence rates of GBS or *E. coli* EOS). We analyzed all cases of GBS and *E. coli* culture-proven neonatal sepsis that have been entered into the network database, coming from intra- or extra-regional birthing centers. 

Case reporting and isolate collection were determined to be an active surveillance of public health interest. The Ethical Committee of the coordinating center (Azienda Ospedaliero-Universitaria Policlinico di Modena) and of the collaborating centers approved the project (Prot. 910/2020). To maintain patient confidentiality, spreadsheets submitted to the principal investigator were fully anonymous and did not include any identifiable data of patients or caregivers.

### 2.2. Definitions

EOS case: positive blood or CSF culture, obtained within the third day of life [22,23].LOS case: positive blood or CSF culture obtained after the 3rd and within the 90th day of life [22,23].Preterm neonates: neonates born at less than 37 (<37) weeks’ gestation.Very-low-birth-weight (VLBW) neonates: neonates with a birth weight under 1500 (<1500) grams.Asymptomatic neonate: infant without any symptoms of sepsis, with positive blood culture collected because of the presence of maternal risk factors.Meningitis: clinical signs and CSF positive culture and/or positive polymerase chain reaction (PCR) [1,5,24].Disease severity: mild, moderate, or severe disease according to the clinical judgment of the attending physician.

### 2.3. Statistical Analysis

Statistical analyses were performed using Stata Direct Statistical Software version 13 (StataCorp LP, College Station, TX, USA). Continuous variables were reported as median and interquartile range (IQR), while categorical variables were reported using frequencies. The groups were compared by χ^2^ analyses for categorical variables and by Mann–Whitney rank sum test or Kruskal–Wallis tests as appropriate as non-parametric tests for continuous variables. Rate of LP and related factors were assessed at uni- and multivariate logistic regression analysis. We used an exploratory strategy for including variables in the multivariable model. A *p*-value < 0.05 was considered significant. McNemar’s test was used for the comparison of positive CSF by culture or PCR.

## 3. Results

During the study period, 400 neonates were diagnosed with EOS or LOS; EOS was confirmed in 144 neonates (GBS, *n* = 89; *E. coli*, *n* = 55) and LOS in 256 infants (GBS, *n* = 162; *E. coli*, *n* = 94). Among 400 infants, 335 (83.7%) had sepsis (of which 27 were asymptomatic at the time of blood culture), 60 (15.0%) had both sepsis and meningitis, and 5 (1.3%) had meningitis only. Blood culture was unavailable in five infants who were diagnosed with meningitis. 

The median gestational age at birth of the entire population was 38 weeks and most infants were born full-term (59.9%); the median post-conceptional age at sepsis was 40 weeks (IQR 35–43 weeks). Infants with LOS were more likely to be male and twin; rates of GBS or *E. coli* did not differ between EOS and LOS (Table 1). 

### 3.1. Performing an LP

LP was performed in 228 of 400 (57.0%) infants, and 65 (28.5%) of them had meningitis (Figure 1). LP was performed more frequently in level 3 as compared to level 1–2 centers (205 of 348, 58.9% vs. 22 of 52, 42.3%, respectively, *p* = 0.024). Rates of LP were higher among infants with GBS than in infants with *E. coli* infections; in preterm or VLBW infants, rates of LP were lower in infants with EOS than in those with LOS. Furthermore, concerning the etiology of infection, rates were lower in infants with *E. coli* than in those with GBS infection (Table 2).

Preterm infants who underwent LP had more days of life and higher post-conceptional age at the time of procedure as compared to those who did not undergo LP (median 26.5 days of life, IQR 5.5–41.5 vs. median 6.5 days of life, IQR 0–24, *p* < 0.001; median 35 weeks, IQR 31–37 vs. median 31 weeks, IQR 26–34, *p* < 0.001, respectively). Furthermore, almost 15.3% (61 out of 400) developed septic shock or multi-organ failure; neonates with mild, moderate, or severe disease were 168 (42.0%), 127 (31.8%), and 105 (26.2%), respectively. LPs were more likely to be performed in more severely ill patients as compared to those mildly-to-moderately ill (69 out of 105, 65.7% vs. 159 out of 295, 53.9%, *p* = 0.036, OR 1.28, CI 1.01–1.61). Table 3 shows the uni- and multivariate analysis of variables associated with the procedure. GBS culture positivity and disease severity were both associated with higher rates of LP. 

### 3.2. CSF Parameters, Culture, Polymerase Chain Reaction, and Antibiotic Administration

Among 65 confirmed cases, meningitis was only culture-proven (*n* = 36), only confirmed through PCR (*n* = 17), or both (*n* = 12). LP was performed after starting an antibiotic therapy in 123 (53.9%) infants (median time 12 h after antibiotic initiation, IQR 24–48 h). Rates of culture-proven meningitis were higher when LP was performed before starting antibiotics (35 out of 48 cases prior to antibiotics, 72.9%, vs. 67 out of 173 after antibiotics, 38.7%, *p* < 0.001). The probability of obtaining a positive CSF culture was lower if the LP was performed more than 12 h after the start of antibiotics (>12 h, 4/79, 5.1%, vs. <12 h, 9/40, 22.5%, *p* = 0.004). The analysis of 55 infants who underwent both CSF culture and PCR showed that the probability of having a positive CSF culture was lower than the probability of having CSF PCR positive when antibiotics had been already administered at the time of the procedure (6/55, 10.9%, positive CSF culture before antibiotics, vs. 19/55, 34.6%, positive CSF PCR after the start of antibiotics, absolute difference 23.7%, 95% CI 9.5–37.7%, *p* = 0.001). 

Among 228 LPs, 93 (40.8%) were contaminated by blood. The comparison of CSF parameters (protein, glucose, and number of cells) among the remaining LPs found significant differences between 42 infants with positive PCR or culture versus 93 infants with sterile CSF and/or negative PCR. No differences were found between CSF parameters among infants with positive CSF (culture and/or PCR) when LPs were performed prior to or after the initiation of antibiotics (Table 4).

### 3.3. Use of Antibiotics and Outcome according to LP

Infants who underwent an LP were more likely to receive prolonged courses of antibiotics (infants with an LP, median 12 days, IQR 10–15 vs. infants without an LP, median 10 days, IQR 7–14, *p* < 0.001), but were less likely to die (7/228, 3.1% vs. 24/172, 14.0%, *p* < 0.01). Moreover, antibiotic courses were longer in infants with positive CSF as compared to those with sterile CSF (median 17.5 day, IQR 14–21 vs. median 10 days, IQR 10–14, *p* < 0.001).

The use of cephalosporins or carbapenems did not differ in infants undergoing an LP as compared to those who did not undergo an LP (130/228, 43.0% vs. 93/166, 44.0%, *p* = 0.844). However, cephalosporins or carbapenems were less likely to be administered according to the CSF results (infants with negative CSF 53/110, 32.5% vs. infants with positive CSF, 45/65, 69.2%, *p* < 0.001). 

As compared to infants who had no LP, infants undergoing an LP were less likely to die (infants with LP, 7/228, 3.1% vs. infants without an LP, 24/172, 14.0%, *p* < 0.001). By logistic multivariate logistic regression analysis (including LP performance and the severity of the disease), failure to perform an LP was independently predictive of death (*p* < 0.001, OR 0.83, 95% CI 0.03–0.22, by logistic regression analysis). 

## 4. Discussion

In this prospective multicenter study, just over half of infants (57%) underwent LP. Missed LPs could be even higher in clinical practice, since our infants had culture-proven infection, a condition in which LP should be performed [7]. The rate of LP performed during sepsis found in the current study is consistent with the literature, which reports a wide range (from 30% to 70%) in infants with EOS [15,16,19]. Unfortunately, the currently available blood tests are too inaccurate to exclude bacterial meningitis without performing an LP [7]. Although LP should be postponed in hemodynamically unstable neonates, antibiotic treatment should be started soon after blood culture [14]; withholding LP may lead either to failure in administering effective antibiotic therapy (antibiotics that do not pass the blood–brain barrier) or to exposing infants without meningitis to unnecessary antibiotics [25]. Neither the association of LP with a high failure rate, nor the potential complications associated with the procedure should discourage clinicians from performing an LP, given that it has been demonstrated that LP is usually safe and that potential events are mostly very manageable in the hospital setting [8]. 

Premature neonates with sepsis bear the highest risks of developing meningitis [26], but we found that premature infants rarely underwent LP (especially during the first days of life). This finding aligns with current literature. Stoll et al. showed that neonates under 28 weeks’ gestational age and/or VLBW neonates during sepsis undergo LP in less than half of the cases compared to more than 80% of full-term infants [15,16]. Other recent studies report a low performance in preterm or VLBW neonates (9.3% of infants with body weight < 1000g; only 2% of infants with gestational age <32 weeks and infants with body weight < 1000g were excluded in the study protocol) [8,9]. Another key issue is the timing of LP in relation to the initiation of antibiotic therapy. In our study, most LPs (53.9%) were performed after the start of antibiotics, with a median delay of 12 h. In the literature, this percentage is reported to be up to 95.5% [16]. We speculate that this delay may be related to clinicians’ habit of waiting for blood culture positivity before performing LP. However, it is well-known [27,28] that up to 30% of meningitis cases are associated with negative blood culture. Another hypothesis is that some physicians perform the LP only when clinical conditions are worse. Indeed, in our study, the rate of LP was higher in infants with the highest severity of the disease. Nevertheless, given the non-specific clinical signs of meningitis in infants, withholding LP in infants with mild and moderate symptoms may lead to underdiagnosing meningitis and delaying the start of effective antibiotic therapy. In fact, performing an LP after antibiotics administration is associated with a reduced sensitivity of CSF analysis. Both CSF parameters (protein and glucose levels, but not cellularity) and CSF culture are affected by the administration of antibiotics [29]. In our study, CSF parameters (protein, glucose, and cells) in infants with meningitis were similar when LP was performed before or after antibiotics initiation. By contrast, rates of culture-proven meningitis were higher when LP was performed prior to the initiation of antibiotics. Similar to Nigrovic et al. [29], we found that children treated with antibiotics >12 h before LP had lower rates of positive culture (84 vs. 58%). Other investigators reported that the timing of CSF sterilization varied according to the pathogen (15 min to 2 h for *N. meningitidis*, 4 h for *S. pneumoniae,* and 8 h for group B strep) [30]. 

In such cases, the sensitivity of LP is increased through the use of CSF PCR, which is unaffected by antibiotics. Indeed, in our study, CSF PCR results were more frequently positive compared to CSF culture [24,31,32]. Even if the PCR technique is particularly valuable, it also has some limitations; it is not available in all centers (especially in low-income countries), it is not guaranteed to include all potentially involved pathogens, and it is particularly expensive. 

Finally, in our study, performing LP was confirmed to be an important guide for antibiotic therapy. Indeed, performing an LP was predictive of higher survival (although failure to perform an LP could depend on extreme clinical severity of some neonates) and the duration and choice of antibiotics was targeted according to CSF results (infants with positive CSF received longer courses of antibiotics, being more frequently cephalosporin and carbapenem). 

The strengths of this study are the prospective multicentric design, the large cohort of septic infants evaluated, the inclusion of very recent data, and the evaluation of factors associated with the rate of performing LP, previously poorly investigated. However, this study has potential limitations. First, we did not include neonates with CSF pleocytosis, sterile cultures, and negative PCR; although this condition is uncommon, some cases of meningitis may have been missed. Second, we did not use a specific laboratory cut-off to define bloody CSF, but we referred to the notes of the laboratory or the personal judgment of physicians performing the LP. Third, because the use of PCR has spread only in recent years, data regarding CSF PCR were unavailable in more than half of LPs, and this potentially limits the reliability of the comparison between CSF culture and CSF PCR positivity. Furthermore, we could not exclude that some unsuccessful LPs (because of bloody or unavailable CSF) have not been reported by physicians, underestimating the real rate of LP in the population. An additional limitation is that the period of evaluation of infants with GBS infection (from 2003 to 2022) was longer in comparison with that of infants with *E. coli* infection (from 2016 to 2022). This would be a bias, as the clinical practice related to the performance of LP may have changed over time. Because infants with GBS or *E. coli* infection were enrolled in the study, the generalization of the results to infections caused by additional pathogens should be undertaken with caution. Finally, we did not collect information regarding reasons that led physicians not to perform or delay the LP.

## 5. Conclusions

We emphasize the importance of performing an LP to rule out meningitis and to administer the most appropriate antimicrobial therapy during sepsis. LP is even more necessary in premature infants who have the highest risks of meningitis, mortality, and long-term complications.

## Figures and Tables

**Figure 1 microorganisms-11-01546-f001:**
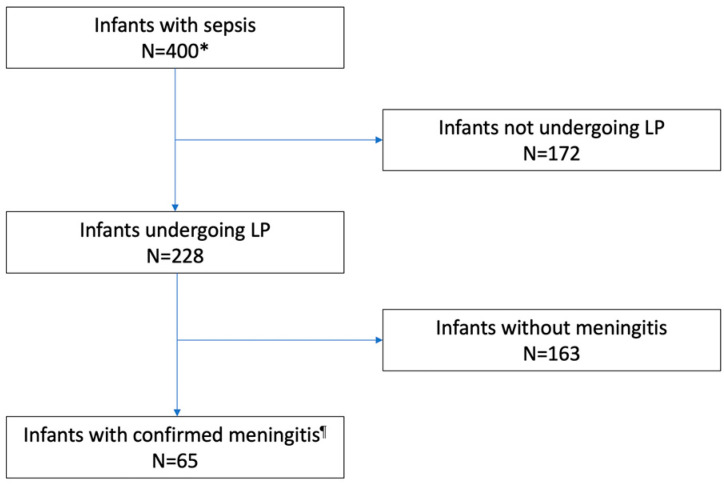
Flow diagram: performing LP and CSF positivity. LP, lumbar puncture. * Blood culture was unavailable in 5 infants who were diagnosed with meningitis. ^¶^ Culture or PCR Positive CSF.

**Table 1 microorganisms-11-01546-t001:** Demographics of infants enrolled in the study.

Variables	All Infants *N* = 400	EOS *N* = 144	LOS *N* = 256	*p*
Pathogen, *N* (%)				
GBS	251 (62.7)	89 (61.8)	162 (63.3)	0.770
*E. coli*	149 (37.3)	55 (38.2)	94 (36.7)	
GA, wks (IQR)	38 (31–40)	38 (32–40)	38 (31–40)	0.356
Preterm delivery, *N* (%)	156 (39.1)	52 (36.1)	104 (40.8)	0.358
Body weight at birth, g (IQR)	2887 (1476–3415)	2932 (1517–3430)	2877 (1475–3410)	0.661
Very-low-birth-weight neonates, *n* (%)	102 (25.5%)	36 (25.0)	66 (25.8)	0.863
Sex (male), *N* (%)	220 (55)	65 (45.1)	155 (60.5)	0.003
Twins, *N* (%)	42 (10.5)	7 (4.8)	35 (13.7)	0.006
Mode of delivery, *n* (%)				
Vaginal	249 (62.3)	95 (66)	154 (60.2)	0.249
Cesarean Section	151 (37.7)	49 (34)	102 (39.8)	

EOS, early onset sepsis; GA, gestational age at birth; GBS, group B *streptococcus*; LOS, late-onset sepsis; wks, weeks.

**Table 2 microorganisms-11-01546-t002:** LP performance and comparison between groups (EOS vs. LOS and GBS vs. *E. coli*).

Variables	All *N* = 400	EOS *N* = 144	LOS *N* = 256	*p*	GBS *N* = 251	*E. coli* *N* = 149	*p*
LP performed, *N* (%)	228 (57)	77 (53.5)	151 (59.0)	0.285	179 (71.3)	49 (32.9)	<0.001
LP performed in preterm infants, *N* (%) ^†^	84 (53.9)	19 (36.6)	65 (62.5)	0.002	66 (72.5)	18 (27.7)	<0.001
LP in VLBW neonates, *n* (%) ^¶^	50 (49)	11 (30.6)	39 (59.1)	0.006	38 (65.5)	12 (27.3)	<0.001
LP performed after initiation of antibiotics, *N* (%) ^§^	123 (53.9)	55 (71.4)	68 (45.0)	<0.001	94 (52.8)	28 (57.1)	0.590
Hours lasting from initiation of antibiotics to LP, median (IQR)	12 (24–48)	24 (20–48)	16 (6–36)	0.014	24 (12–48)	22 (8–34)	0.438

EOS, early-onset sepsis; GBS, group B *streptococcus*; LOS, late-onset sepsis; LP, lumbar puncture. One infant with LOS by *E. coli* had spinal malformation (terminal filum anchorage) and did not undergo LP. ^†^ The percentage was calculated on the number of preterm infants (total *n* = 156; EOS, *n* = 52; LOS, *n* = 104). ^¶^ The percentage was calculated on the number of VLBW infants (total *n* = 102; EOS, *n* = 36; LOS, *n* = 66). ^§^ The percentage was calculated on the number of infants undergoing an LP (total *n* = 228; EOS, *n* = 77; LOS, *n* = 151; GBS, *n* = 179, *E. coli*, *n* = 49).

**Table 3 microorganisms-11-01546-t003:** Univariate and multivariate analysis of factors associated with LP performance.

Variables	Univariate			Multivariate		
	OR	95% CI	*p*	OR	95% CI	*p*
Pathogen (GBS vs. *E. coli*)	0.197	0.12–0.30	<0.001	0.20	0.13–0.32	<0.001
Preterm delivery	0.815	0.54–1.22	0.325	-	-	-
Birth weight	1.00	0.99–1.00	0.153	-	-	-
Higher disease severity	1.435	1.11–1.84	0.005	1.34	1.02–1.77	0.031
Center level	0.505	0.28–0.91	0.023	0.72	0.37–1.39	0.335
EOS vs. LOS	1.25	0.82–1.88	0.285	-	-	-
Postmenstrual age	1.02	0.99–1.05	0.078	-	-	-
Days of life at the sepsis	1.00	0.99–1.00	0.756	-	-	-

EOS, early-onset sepsis; GBS, group B *streptococcus*; LOS, late-onset sepsis.

**Table 4 microorganisms-11-01546-t004:** CSF parameters and comparison between groups (93 bloody LPs were excluded from the analysis).

Variables	Culture or PCR Positive CSF (*N* = 42)	Sterile CSF (*N* = 93)	Missing	*p*	CSF Prior to Antibiotic Initiation in Positive PCR/Culture (*N* = 21)	CSF after Antibiotic Initiation in Positive PCR/Culture (*N* = 18)	Missing	*p*
Proteins								
Median (IQR)	290 (198–500)	80 (52–115)	10	<0.001	292 (216–500)	243 (197–463)	3	0.639
Glucose								
Median (IQR)	19 (9–35)	51.5 (44.5–64)	13	<0.001	19 (9–32)	23 (15–35)	3	0.354
Number of cells								
Median (IQR)	345.5 (19–884.5)	4 (2–11)	13	<0.001	500 (8–869)	341 (53–1660)	3	0.536

PCR, polymerase chain reaction. IQR, interquartile range.

## Data Availability

Data are available upon request to the corresponding author.

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
