# Peer review of "Lumbar Puncture and Meningitis in Infants with Proven Early- or Late-Onset Sepsis: An Italian Prospective Multicenter Observational Study"

_microorganisms, 2023, doi:10.3390/microorganisms11061546_

Round 1
Reviewer 1 Report
1) Brief Summary
This work investigates rates of lumbar puncture (LP) performance in a large cohort of 102 infants with early- (EOS) or late-onset sepsis (LOS) due to group B streptococcus (GBS) or 103 E. coli, as well as variables associated with LP performance. The manuscript is well written, and the data is relevant; however, a few changes are necessary to improve the merit of this paper further.
2) General concept comments
I. The Introduction is clear and appropriate. Nevertheless, the first paragraph should be divided into two new ones. The first paragraph would end at line 83 after symptoms are often subtle." The second would start with, "Although CSF analysis and culture are essential for diagnosing meningitis in infants...". This separation would facilitate the reader's comprehension.
II. The Materials and Methods.
My primary remark regarding this manuscript is related to the methods. In lines 107-108, the authors state, "This was a prospective observational study led from January 1st, 2003 to December 31st, 2022". Later, it is mentioned that "The surveillance network initially concerned the active area-based surveillance of GBS infections [18]. Since 2016, E. coli surveillance has been 115 added..." (lines 114-116). My questions to the authors are as follows:
II.a) Have you included GBS infection data obtained before E. coli surveillance (2003-2016) in this research, as indicated in lines 107-108?
II.b) If so, you have compared GBS infection numbers from 2003 to 2022 with E. coli data from 2016 to 2022. Do you stand behind this comparison? Wouldn't it be more correct to present the GBS infection numbers from 2003 to 2016 and compare them to E. coli data only from 2016 to 2022? Please elaborate.
III. Results
The presentation of the results is appropriate, and the tables are well-designed and self-explanatory.
My only observation is related to the data collection period of GBS vs. E. coli, as mentioned above. For instance, in lines 178-179, the authors report that "Rates of LP were higher among infants with GBS than in infants with E. coli infections." At first glance, this observation would be explained as a higher incidence of GBS than E. coli, a significant result. However, it could also be a clerical error, i.e., a methodological flaw, explainable by the fact that GBS case numbers have been obtained since 2003, while E. coli reports were only available from 2016. So, it is essential to ensure that the cases compared were collected from the same period.
IV. Discussion
The discussion is clear and addresses the main results. The very positive point is that the authors employ their experience to emphasize their points of view on practical approaches based on their results.
V. Conclusion
I have two suggestions for the conclusion.
First, I would remove the sentence: "In this large study, we found a low rate of LP and a high rate of antibiotics administration before performing LP, in the course of suspected or proven neonatal sepsis," which is a summary of results and not a conclusion.
Second, I emphasize the main conclusion of this study. The Conclusion of the manuscript should start with "Since most premature infants have the highest risks of meningitis, mortality, and long-term complications associated with the early bacterial onset and late-onset sepsis, we emphasize the importance of performing the LP, to rule out meningitis in the course of sepsis, especially when the baby is preterm of VLBW infant."
It is necessary to check for minor English Language errors (word spelling, grammar).
Author Response
We warmly thank the referee for the valuable suggestions, and certainly our manuscript will be improved after changes. Below we reply to each point.
I) "The first paragraph would end at line 83 after symptoms are often subtle." The second would start with, "Although CSF analysis and culture are essential for diagnosing meningitis in infants...". This separation would facilitate the reader's comprehension."
Answer: We separated the paragraph as suggested.
II.a) Have you included GBS infection data obtained before E. coli surveillance (2003-2016) was started in this research, as indicated in lines 107-108?
Answer: Yes we have included GBS infection data obtained before E. coli surveillance (2003-2016). We modified as follows: "Since 2003 the surveillance network concerned the active area-based surveillance of GBS infections [18], and since 2016 E. coli surveillance has been added [21]".
II.b) If so, you have compared GBS infection numbers from 2003 to 2022 with E. coli data from 2016 to 2022. Do you stand behind this comparison? Wouldn't it be more correct to present the GBS infection numbers from 2003 to 2016 and compare them to E. coli data only from 2016 to 2022? Please elaborate.
Answer: We confirmed that we performed the statistical evaluations regarding all GBS infections (from 2003 to 2022) and all E. coli infections (from 2016 to 2022). Presenting the GBS infection numbers from 2003 to 2016 and comparing them to E. coli data only from 2016 to 2022 would exclude a high proportion of infants with GBS in the period 2016 to 2022 and we do not think it would be completely correct. However, we agree that possible changes in clinical practice may have occurred during time, given the long period of study, and that the confidence in performing the LP may have changed over time. For this reason we modify the limits of the study as follows: "An additional limitation of the study is that the period of evaluation of infants with GBS infection (from 2003 to 2022) was longer in comparison with that of infants with E. coli infection (from 2016 to 2022), this may constitute a bias as the clinical practice related to the performance of LP may have changed over time."
III) My only observation is related to the data collection period of GBS vs. E. coli, as mentioned above. For instance, in lines 178-179, the authors report that "Rates of LP were higher among infants with GBS than in infants with E. coli infections." At first glance, this observation would be explained as a higher incidence of GBS than E. coli, a significant result. However, it could also be a clerical error, i.e., a methodological flaw, explainable by the fact that GBS case numbers have been obtained since 2003, while E. coli reports were only available from 2016. So, it is essential to ensure that the cases compared were collected from the same period.
Answer: Regarding the given example (Rates of LP were higher among infants with GBS than in infants with E. coli infections), we underline that we evaluated the proportion of infants who underwent a LP, which does not necessary depend on the number of infants evaluated, but by the propensity of physician to perform a LP. The higher rates of LP in infants with GBS compared to infants with E. coli does not depend on the number of infants with GBS or E.coli infections but on the number of LP performed in each group (rates). Given this consideration we do not think that the different number of infants for each group (GBS or E. coli) could have affected the results regarding the performance of LP. However, we mentioned the difference in the period of evaluation in limitations, as specified above. In addition, after your consideration we performed a further evaluation (not reported in the manuscript) including infants born after 2016 only and results regarding the comparison between infants with GBS and E. coli infection remained significant. Moreover, as specified in methods, as the surveillance network was not entirely area-based, the study had no epidemiologic purpose (i.e., incidence rates of GBS or E. coli).
V) I have two suggestions for the conclusion. First, I would remove the sentence: "In this large study, we found a low rate of LP and a high rate of antibiotics administration before performing LP, in the course of suspected or proven neonatal sepsis," which is a summary of results and not a conclusion. Second, I emphasize the main conclusion of this study. The Conclusion of the manuscript should start with "Since most premature infants have the highest risks of meningitis, mortality, and long-term complications associated with the early bacterial onset and late-onset sepsis, we emphasize the importance of performing the LP, to rule out meningitis in the course of sepsis, especially when the baby is preterm of VLBW infant."
Answer: we modified conclusions as follows: "We emphasize the importance of performing an LP to rule out meningitis and to administer the most appropriate antimicrobial therapy during sepsis. LP is even more necessary in premature infants who have the highest risks of meningitis, mortality, and long-term complications".
"
VI) It is necessary to check for minor English Language errors (word spelling, grammar).
Answer: we checked and corrected English Language errors.
We are very grateful for the review.
Sincerely
Reviewer 2 Report
I read with interest the paper entitled “Propensity to perform a lumbar puncture in early or late-onset sepsis: an Italian prospective multicenter study”.
The authors performed a multicenter study to evaluate the rates of performing lumbar puncture and its possible clinical significance in infants with bacteriaemia (E.coli or GBS).
This manuscript is well written. The authors address an interesting research question.
However, there are several issues that authors should address to improve the manuscript:
1.) Why only GBS and E.coli were included? How this inclusion criteria limits the generalization of results?
2.) Please include a flow diagram of the data, how it was selected and the different losses to end up with your final sample (e.g. CONSORT style flow diagram).
3.) Please describe what type of data was missing and how this was managed in the analysis. This could also form part of the flow diagram.
4.) The Results section “3.3. Use of antibiotics and outcome according to LP” is interesting and should be expanded. The authors suggest that infants with LP had better outcomes. I would suggest adding the Table with more data. Furthermore, it is not adequately described if the associations presented between the outcome and the different risk factors are independent of each other or if there is some level of correlation. This full model for estimating the association need to be fully described.
5.) The title is hard to read and unclear. Consider rephrasing it.
The manuscript require only minor English editing. Consider to change the title.
Author Response
We warmly thank the referee for the valuable suggestions, and certainly our manuscript will be improved after changes. Below we reply to each point.
1.) Why only GBS and E.coli were included? How this inclusion criteria limits the generalization of results?
Answer: The surveillance network was started in 2003 as surveillance of GBS infections and since 2016 E. coli surveillance was added. For this reason we included only GBS and E. coli. We specified in the limits that the generalization of the results to infants with infections caused by additional pathogens should be cautious.
2.) Please include a flow diagram of the data, how it was selected and the different losses to end up with your final sample (e.g. CONSORT style flow diagram).
Answer: In the network infants with infections by GBS or E. coli only were included, no infants were excluded from the evaluation. The final sample overlap with the initial population. We added the flow diagram.
3.) Please describe what type of data was missing and how this was managed in the analysis. This could also form part of the flow diagram.
Answer: when missing data was present the number was specified (table 4). When a certain data was missed the corresponding case was excluded for that specific assessment.
4.) The Results section “3.3. Use of antibiotics and outcome according to LP” is interesting and should be expanded. The authors suggest that infants with LP had better outcomes. I would suggest adding the Table with more data. Furthermore, it is not adequately described if the associations presented between the outcome and the different risk factors are independent of each other or if there is some level of correlation. This full model for estimating the association need to be fully described.
Answer: we completely agree that this topic is interesting and particularly relevant, and we were excited about such results. However, given that this was not the main aim of this study and that currently we do not have further details regarding the outcome of the patients involved, we have not been able to explore this issue further. It will certainly be the topic of an upcoming research. Regarding the details of the regression model, we modify as follows: "By logistic multivariate logistic regression analysis (including LP performance and the severity of the disease), failure to perform an LP was independently predictive of death (p<0.001, OR 0.83, 95% CI 0.03-0.22, by logistic regression analysis)."
5) The title is hard to read and unclear. Consider rephrasing it.
Answer: we changed the title as follows "Lumbar puncture and meningitis in infants with proven early or late-onset sepsis: an Italian prospective multicenter observational study"
We are very grateful for the review.
Sincerely